# Trajectory Planner for UAVs Based on Potential Field Obtained by a Kinodynamic Gene Regulation Network

**DOI:** 10.3390/s23187982

**Published:** 2023-09-20

**Authors:** Juncao Hong, Diquan Chen, Wenji Li, Zhun Fan

**Affiliations:** 1College of Engineering, Shantou University, Shantou 515063, China; 2Key Laboratory of Intelligent Manufacturing Technology, Shantou University, Ministry of Education, Shantou 515063, China; 3International Cooperation Base of Evolutionary Intelligence and Robotics, Shantou University, Shantou 515063, China

**Keywords:** trajectory planning, kinodynamic constraints, environmental perception, potential field, gene regulation network

## Abstract

Quadrotor unmanned aerial vehicles (UAVs) often encounter intricate environmental and dynamic limitations in real-world applications, underscoring the significance of proficient trajectory planning for ensuring both safety and efficiency during flights. To tackle this challenge, we introduce an innovative approach that harmonizes sophisticated environmental insights with the dynamic state of a UAV within a potential field framework. Our proposition entails a quadrotor trajectory planner grounded in a kinodynamic gene regulation network potential field. The pivotal contribution of this study lies in the amalgamation of environmental perceptions and kinodynamic constraints within a newly devised gene regulation network (GRN) potential field. By enhancing the gene regulation network model, the potential field becomes adaptable to the UAV’s dynamic conditions and its surroundings, thereby extending the GRN into a kinodynamic GRN (K-GRN). The trajectory planner excels at charting courses that guide the quadrotor UAV through intricate environments while taking dynamic constraints into account. The amalgamation of environmental insights and kinodynamic constraints within the potential field framework bolsters the adaptability and stability of the generated trajectories. Empirical results substantiate the efficacy of our proposed methodology.

## 1. Introduction

The successful implementation of autonomous drone navigation has immense value in scientific studies and various industries as it liberates their applications from limitations and paves the way for diverse deployments. Despite this progress, the pursuit of safe and efficient navigation in unknown environments poses significant challenges. To achieve secure and efficient navigation, it is of paramount importance for drones to accurately represent and effectively utilize environmental information during trajectory planning.

A widely adopted technique for representing the distance between a quadrotor and objects in its environment is the Euclidean signed distance field (ESDF), which utilizes gradients. The ESDF has proven to be effective in motion planning for single-agent scenarios, providing valuable insights for autonomous navigation.

Nevertheless, it has limitations in certain application scenarios. For instance, its ability to represent all objects in an environment as obstacles may not adequately capture the complexity of cooperative swarm motions or environments with multiple layers of semantic information. This limitation impedes the successful completion of flight tasks in complex scenarios.

To address these challenges, this study proposed a novel gradient potential field, termed the kino-gene regulatory network (K-GRN) potential field, which is based on a GRN model. By considering the kinematic state of a quadrotor, the objective was to enhance trajectory planning and improve the overall navigation performance. Additionally, we introduced the K-GRN planner, which is a local trajectory planning framework that leverages the kinodynamic GRN potential field to generate optimized flight paths.

Inspired by the intricate interactions observed in organisms, a GRN model was used to generate a gradient potential field. This approach has found extensive application in swarm robot morphology generation and serves as a foundation for the development of autonomous systems. Based on the previous work by Li [1], who utilized the GRN potential field for shape generation and cooperative motion of a swarm of drones without communication, this study further enhanced the potential field by incorporating the kinematic state of the robots. This enhancement enabled the design of a local trajectory planning framework using the proposed potential field, thereby ensuring improved navigation capabilities.

The contributions of this study are as follows. First, we propose an improved GRN potential field that demonstrates its applicability in achieving safe and efficient trajectory planning for drones. Second, we introduce the incorporation of the kinematic characteristics of agents, enabling the extension of the potential field from a spatial domain to a state space, thus, enriching navigation capabilities. Finally, based on these contributions, we establish constraint rules for the trajectory search and develop a comprehensive framework for local planning that further enhances the overall navigation system.

The remainder of this paper is organized as follows. In Section 2, we present a comprehensive review of the related work, encompassing GRN, trajectory planning for unmanned aerial vehicles (UAVs) in positional environments, and the application of bio-inspired algorithms in UAV navigation. Section 3 outlines the enhancements made to the GRN potential field and presents the framework of the proposed method. We present a real-time planning algorithm based on the K-GRN potential field in Section 4. In Section 5, we describe the experimental setup and results for validating the effectiveness of our proposed approach. Finally, in Section 6, we present the conclusions by summarizing the key findings and contributions and provide insights into potential avenues for future research and development.

## 2. Related Work

### 2.1. GRN Model in UAV Control

In recent years, GRNs have attracted considerable attention in the design of control methods for robot control [2,3]. Taylor et al. [4] introduced a GRN–cell adhesion model (CAM) control method that combines a CAM with a GRN model. This approach enables the spatial and temporal differentiation of protein expression across swarm robots, which are treated as artificial cells with virtual membranes and artificial cell adhesion molecules. Guo et al. [5,6] applied a GRN model to generate adaptive patterns for swarm robots. Each robot possessed two genes that produced proteins to control its movements in the x and y directions via GRN mechanisms. These proteins enabled the robots to move within the boundaries of predefined shapes.

Furthermore, evolutionary algorithms are utilized to optimize robot motion parameters. However, this method relies on global coordinates, which are often challenging for robots to obtain in real-world applications. To address this issue, Guo et al. [7] proposed a GRN-based method for selecting a reference robot and establishing a local coordinate system. Each robot contributed to the formation of an uneven B-spline pattern using a local coordinate system. Building on this local coordinate system, Jin et al. [8] introduced a hierarchical gene regulatory network (H-GRN) for adaptive pattern generation, specifically for trapping targets in dynamic environments. Oh et al. [9] expanded an H-GRN model to achieve area coverage. To address the challenges related to merging and separating patterns during multi-target entrapment and obstacle avoidance, Oh et al. [10] proposed an enhanced H-GRN (EH-GRN) structure that incorporates obstacles and targets as inputs and evolves the GRN to generate a morphogen gradient space. Similarly, Meng and Guo [11] introduced an evolving GRN method to adjust the structure and parameters of the GRN.

Recently, Wu et al. [12] proposed a collaborative GRN (C-GRN) model that enables peer collaboration between agents. This model enables agents to discover and reinforce weak areas in the formed patterns. Yuan et al. [13] combined a tracking-based H-GRN with a leader–follower model and introduced a TH-GRN model that is suitable for dynamic and complex environments. Fan et al. [14] used a genetic programming approach to design a design automation framework for the gene regulatory network.

In summary, the utilization of a GRN is widespread for generating gradient fields that depict environmental information and direct robot movement. However, the current studies either rely solely on gradient information to guide robot behavior or address a GRN method as a distinct upper-level structure, separate from a lower-level planning controller.

The proposed GRN structure is based on VG-Swarm [1] improvement, which expands the dimensions of the GRN potential field and incorporates the state space of drones. We designed a real-time trajectory-planning method specifically tailored to this framework.

### 2.2. Bio-Inspired Path Planning

Bio-inspired algorithms have recently attracted substantial attention owing to their efficacy in solving complex optimization problems while maintaining balance among their components. In the domain of path planning problems, studies have increasingly focused on harnessing the power of bio-inspired algorithms to optimize these problems [15,16,17,18].

Swarm intelligence is derived from the self-organizational behavior observed in biological systems inspired by this swarm intelligence, we have developed a number of swarm intelligence algorithms, among which the most notable are particle swarm optimization (PSO) and ant colony optimization (ACO). For example, Phung et al. [19] proposed an enhanced discrete PSO algorithm tailored to UAV inspection path planning. They replaced the inspection path-planning problem with a solution derived from the extended traveling salesman problem. ACO, originally introduced by Dorigo et al. in 1999 [20], enables ants to cooperatively discover the shortest path between a nest and its food sources using pheromone guidance. Nonetheless, both the ACO and PSO algorithms encounter challenges related to slow convergence speeds when the computational complexity rapidly increases, which is undesirable for real-time planning in complex environments.

Another noteworthy bio-inspired algorithm is the evolutionary algorithm. Sun et al. employed an adaptive multi-objective differential evolution (DE) algorithm in a multi-robot system [21]. Patlea et al. presented an improved genetic algorithm (GA) that employed a binary code matrix for mobile robot navigation in complex environments [22]. However, similar to the ACO and PSO algorithms, the time complexity of evolutionary algorithms is typically high, posing obstacles to achieving real-time performance in complex environments.

One limitation of the bio-inspired algorithms is their lack of effectiveness in real-time path-planning problems. This is because of the requirement for the learning processes, such as the modification of path selection probabilities using an objective function, as observed in ant colony optimization. Among these bio-inspired methods, a GRN model offers a superior option because of its capability to generate gradient fields in real-time. For the first time, we introduced a real-time trajectory planning framework using a potential field generated from a GRN model.

### 2.3. UAVs Navigation with Gradient Field

Gradient-based motion planning is the dominant approach for generating local trajectories in UAVs, formulating the motion planning as an unconstrained nonlinear optimization problem. The introduction of the Euclidean signed distance field (ESDF) in robotic motion planning by Ratliff et al. [23] has enabled planning frameworks to directly optimize trajectories in a configuration space using its abundant gradient information. However, discrete-time trajectory optimization [23,24] is unsuitable for drones because of its sensitivity to dynamic constraints. To address this limitation, a continuous-time polynomial trajectory optimization method for UAV planning was proposed by [25]. Despite its merits, the method carries a significant computational burden due to the integration of the potential function, and even with random restarts, the success rate of this method approximates only 70%. To address these challenges, Ratliff et al. [23] introduced a B-spline parameterization of trajectories leveraging convex hull properties. In [26], the success rate was significantly improved by finding a collision-free initial path as the front end, and further improvements were made by considering the kinodynamic constraints [27]. Zhou et al. [28] enhanced system robustness by incorporating perception awareness. Among these approaches, the ESDF is crucial in evaluating the distance to nearby obstacles based on the gradient magnitude and direction.

Oleynikova [29] and Han [30] proposed incremental methods for ESDF generation, namely, Voxblox and FIESTA, respectively. Although these methods are highly efficient for dynamic updating, they only provide information on obstacles and neglect other important environmental information in the generated ESDF. Therefore, there is a substantial need to design a gradient field for real-time trajectory planning that can effectively represent environmental information as comprehensively as possible.

## 3. Kinodynamic GRN Potential Field

Our method improves upon the framework of a GRN model in a VG-Swarm [1], as shown in Figure 1. Within this enhanced framework, the method for computing the GRN concentration in each cell is calculated through Equations (1)–(6).
(1)$$\begin{array}{c} \frac{dT_{i}}{dt}=\nabla^{2}T_{i}+\gamma_{i}-T_{i} \end{array}$$
(2)$$\begin{array}{c} \frac{dO_{j}}{dt}=\nabla^{2}O_{j}+\beta_{j}-O_{j} \end{array}$$
(3)$$\begin{array}{c} \frac{dN_{m}}{dt}=\nabla^{2}N_{m}+\eta_{m}-N_{m} \end{array}$$
(4)$$\begin{array}{c} \left\{\begin{array}{c} N=\sum_{m=1}^{n_{n}}N_{m}\\ T=\sum_{i=1}^{n_{t}}T_{i}\\ O=\sum_{j=1}^{n_{o}}O_{j} \end{array}\right. \end{array}$$
(5)$$\begin{array}{c} \frac{dM}{dt}=-M+\mathrm{sig}\left(1-T^{2},\theta ,k\right)+\mathrm{sig}\left(O^{2},\theta ,k\right)\\ +\mathrm{sig}\left(N^{2},\theta ,k\right) \end{array}$$
(6)$$\begin{array}{c} \mathrm{sig}\left(x,\theta ,k\right)=\frac{1}{1+e^{-k(x-\theta )}} \end{array}$$

$\gamma_{i}$, $\beta_{j}$, and $\eta_{m}$ are positive numbers representing the position of the target, obstacles, and neighboring UAVs, respectively. $T_{i}$ represents the protein concentration formed by the $i^{th}$ target in the environment ($\gamma_{i}$ stands for the position information of the $i^{th}$ target), $O_{j}$ represents the protein concentration formed by the $j^{th}$ obstacle ($\beta_{j}$ stands for the position information of the $j^{th}$ obstacle), and $N_{m}$ represents the protein concentration formed by the $m^{th}$ neighboring UAV ($\eta_{m}$ stands for the position information of the $m^{th}$ neighboring UAV). *T*, *O*, and *N* are the combined concentrations produced by all the targets, obstacles, and neighbors, respectively. ∇ is the Laplace operator, which is defined as the second derivative of $T_{i}$, $O_{j}$, and $N_{m}$. The concentration field obtained by calculating *M* is used to generate entrapping patterns.

Equations (1) and (2) can be revised to Equations (7) and (8), respectively, which are crucial operations in the GRN model for generating a gradient field.
(7)$$\begin{array}{c} T_{i}=e^{-td_{tar,i}} \end{array}$$
(8)$$\begin{array}{c} O_{j}=e^{-td_{obs,j}} \end{array}$$
where $d_{tar,i}$ represents the Euclidean distance between the agent and the $i^{th}$ target, and $d_{obs,j}$ represents the distance between the agent and the $j^{th}$ obstacle. Notably, the VG-Swarm considers only the distance between the objects and agents during the guidance control process. To address this limitation, we propose a specialized GRN method for trajectory planning called kinodynamic GRN. Unlike the VG-Swarm, the kinodynamic GRN introduces the concept of the state transition cost. We assume that any point within the grid can serve as an intermediate node for UAV movement. By sampling the states of these intermediate nodes, denoted by $X_{k+1}\;\left(p,v\right)$, we can calculate the state transition costs to the obstacle and target nodes, denoted by $L_{X_{k+1},X_{obs}}$ and $L_{X_{k+1},X_{tar}}$, respectively. These costs can be expressed using Equations (9) and (10).
(9)$$\begin{array}{cc} \; & L_{x_{k+1},x_{\mathrm{tar}\;}}=\int_{0}^{\tau_{k+1,tar}}∥u_{k+1,\;\mathrm{tar}\;}∥dt+\rho \tau_{k+1,\;\mathrm{tar}\;} \end{array}$$
(10)$$\begin{array}{cc} \; & L_{x_{k+1},x_{obs}}=\int_{0}^{\tau_{k+1,obs}}∥u_{k+1,obs}∥dt+\rho \tau_{k+1,obs} \end{array}$$
where $u_{k+1,tar}$ and $\tau_{k+1,tar}$, respectively, represent the optimal input and optimal time required to transition from the state $X_{k+1}$ to the target state. They are determined using the Pontryagin minimum principle, as expressed in Equation (Equation 11).
(11)$$\begin{array}{c} \left\{\begin{array}{cc} p_{\mu }^{*}\left(t\right) & =\frac{1}{6}\alpha_{\mu }t^{3}+\frac{1}{2}\beta_{\mu }t^{2}+v_{\mu c}+p_{\mu c}\\ \left[\begin{array}{c} \alpha_{\mu }\\ \beta_{\mu } \end{array}\right] & =\frac{1}{T^{3}}\left[\begin{array}{cc} -12 & 6T\\ 6T & -2T^{2} \end{array}\right]\left[\begin{array}{c} p_{\mu g}-p_{\mu c}-v_{\mu c}T\\ v_{\mu g}-v_{\mu c} \end{array}\right]\\ J^{*}\left(T\right) & =\sum_{\mu \in \{x,y,z\}}\left(\frac{1}{3}\alpha_{\mu }^{2}T^{3}+\alpha_{\mu }\beta_{\mu }T^{2}+\beta_{\mu }^{2}T\right) \end{array}\right. \end{array}$$

Next, the Euclidean distance in Equations (7) and (8) is replaced by the state transition cost, as expressed by Equations (9) and (10), thereby forming the protein concentration expressions (Equations (12) and (13)) for environmental information in the kinodynamic GRN potential field.
(12)$$\begin{array}{cc} \; & T_{i}=e^{-tL_{x_{k+1}},x_{tar}} \end{array}$$
(13)$$\begin{array}{cc} \; & \;O_{j}=e^{-tL_{x_{k+1},x_{obs}}} \end{array}$$

In the path-planning process, we also consider the current state of the agent to ensure that the planned path is executable. Hence, the current kinematic state of an individual drone, denoted as $X_{k}$, is considered. We define the forward cost, $L_{X_{k},X_{k+1}}$, using Equation (Equation 14), which represents the cost of transitioning from state $X_{k}$ to $X_{k+1}$.
(14)$$\begin{array}{cc} \; & L_{x_{k},x_{k+1}}=\int_{0}^{\tau_{k,k+1}}∥u_{k,k+1}∥dt+\rho \tau_{k,k+1}\\ \; \end{array}$$

Finally, the generation of the potential field is expressed using Equations (15)–(19), where $M_{k+1}$ denotes the protein concentration in the sampled state $X_{k+1}$.
(15)$$\begin{array}{c} \left\{\begin{array}{cc} \; & T=\sum_{i=1}^{n_{t}}T_{i},\\ \; & O=\sum_{j=1}^{n_{o}}O_{j} \end{array}\right. \end{array}$$
(16)$$\begin{array}{cc} \; & J_{0}=e^{-tL_{x_{k},x_{k+1}}} \end{array}$$
(17)$$\begin{array}{cc} \; & M_{k+1}=\mathrm{sig}\left(1-T^{2},\theta ,k\right)+\mathrm{sig}\left(O^{2},\theta ,k\right)+\mathrm{sig}\left(1-J_{0}^{2},\theta ,k\right) \end{array}$$
(18)$$\begin{array}{cc} \; & \mathrm{sig}\left(x,\theta ,k\right)=\frac{1}{1+e^{-k(x-\theta )}} \end{array}$$

As illustrated in Algorithm 1, during the construction of the K-GRN potential field, the grid map needs to be initialized first. Following that, the grids within *M* are iterated over, during which the state sampling occurs to acquire candidate states. Subsequently, these candidate states within each grid are revisited to compute their associated costs and integrate them into the GRN model. This process ultimately yields concentration information for the specific candidate states at that particular position.
**Algorithm 1** Generation of K-GRN Field**Notation:** Concentration *M*, Environment E cost L State Xe;Initialize:  M←lnitMap(M,X)1:**for** Mi in *M*: **do**2:    **for** Xi in Xk+1: **do**3:        (LXk+1,Xtar)← EvaluateCostToTarget (Xj,E)4:        (LXk+1,Xobs)← EvaluateCostToObstacle (Xj,E)5:        (LXk,Xk+1)← EvaluateCostTolnstance (Xj,E)6:        **if** ¬ isCollsionFree() **then**7:               M(i,j)← CulculateKGRNConcentration (LXk+1,Xtar, LXk+1,Xobs, LXk,Xk+1)8:        **else**9:              SetlnfConcentration (M(i,j))10:        **end if**11:    **end for**12:**end for**13:**return** *M*


In this context, *M* constitutes an assortment of state samples within specific ranges across each grid of the map, indicating the concentration of K-GRN at that position under that particular state. E encompasses environmental data, which include obstacles, objectives, and optionally, neighboring drones. *X* represents the state of the UAV, wherein $X_{k}$ signifies the present state, $X_{k+1}$ denotes the sampled candidate state, and $X_{obs}$ and $X_{tar}$, respectively, refer to the states when the drone encounters an obstacle and when it reaches the target destination. L embodies the cost associated with state transitions, wherein the subscripts of L align with the originating state and the terminal state.

In summary, the K-GRN potential field is a specialized set of potentials crafted for the purpose of trajectory planning. In comparison to the alternative approaches, the potential field molded by K-GRN can embrace a wider array of environmental data, incorporating elements such as obstacles, target points, and neighboring information. It formulates distinct potential fields surrounding these elements to effectively steer the trajectory planning procedure. Furthermore, K-GRN factors in more intricate state information across multiple dimensions, encompassing both the UAV’s current state and the projected future states. This holistic approach ensures the traceability of trajectories and significantly elevates the safety standards for the drone.

## 4. Trajectory Planner Based on K-GRN Potential Field

As mentioned in Section 3, the concentration gradient field of the K-GRN incorporates both environmental information and kinematic considerations to facilitate trajectory planning. However, maintaining a continuous gradient field throughout the planning process can be computationally demanding and impractical, especially for real-time planning in dynamic environments. To address this challenge and improve planner efficiency, we propose a sampling-based approach. Instead of computing the concentration values for all the grid cells, we selectively sample the grid points of interest within the drone’s feasible input range. These sampled points represent potential locations that can be navigated by the drone. By focusing on these points, we can significantly reduce the computational burden and calculate the concentration information efficiently. This approach enables us to strike a balance between computational efficiency and the accurate representation of the environmental information necessary for effective trajectory planning.

A flowchart of the trajectory planner is shown in Figure 2. Initially, a grid map is created in which each grid cell is affected by the protein concentration produced by objects in the environment, including the targets and obstacles. The current input $u_{\tau }$ and its corresponding time τ for the drone are sampled. The cost of the drone executing a trajectory segment is computed using the sampled final state. The path with the lowest cost is selected, and the search process is continued until the vicinity of the target is reached.

## 5. Experiments

The simulation experiments were conducted using UE4 to evaluate the performance of the proposed approach, which can incorporate a wide range of realistic physical models and introduce the various challenges to drone flight. We designed six different scenes, each with two levels of obstacle density, as shown in Figure 3. In each experiment, the obstacles were randomly generated to ensure a diverse and dynamic environment. The navigation task was repeated 100 times for each scene to obtain statistically significant results. In each trial, a random target point within a distance range of 15–20 m was set for the drone. The success or failure of the task was determined based on whether the drone successfully reached the target without colliding with obstacles or experiencing a crash. Through these experiments, we assessed the robustness and effectiveness of the proposed trajectory planner in various challenging scenarios.

To evaluate the effect of different gradient fields on trajectory generation, we conducted experiments to compare various approaches. Specifically, our proposed method was compared against two other approaches: one that utilizes the ESDF-based method, which derives its potential field formation from [30], and another that does not incorporate a potential field at all, as described in [31]. The experimental results are presented in Table 1. In the table, EF refers to the planning method using ESDF, while NF refers to the planning method that does not use potential fields. "Success" is defined as the drone’s capability to traverse from the starting point to the endpoint (within a proximity of roughly 0.3 m) in a jungle setting without colliding with obstacles.

Notably, the proposed method, which integrated both environmental and state information, required a slightly longer computation time than the compared methods. However, the paths generated by the proposed approach strike a balance between safety and efficiency. We observed significant improvements in both the success rate of the trajectory execution and the average flight speed of the drone compared with the benchmark methods. These findings demonstrate the effectiveness of the proposed approach in generating trajectories that consider both safety and efficiency. Despite the slightly longer computation time, the benefits gained in terms of the improved success rate and flight speed render the proposed method a favorable choice for trajectory planning in dynamic environments.

We have also conducted experiments to showcase the efficacy of integrating the dynamics information of drones and their candidate points into the potential field. This underscores the essentiality of evolving from GRN to K-GRN, as presented in Table 2. In the table, KF refers to the planning method that integrates kinematic information into the potential field, while NKF refers to other potential field planning methods that do not include dynamic information. NF is a planning method that does not use potential fields. In the identical scenario and utilizing the same experimental approach, we conducted a comparative analysis of the proposed approach, which involves the K-GRN potential field, against the performance of the GRN potential field without the integration of motion dynamics. Additionally, we contrasted these with navigation task planning methods that do not rely on potential fields. The experimental results underscore a distinct advantage in navigation success rates for the trajectory planning method guided by K-GRN. As previously discussed in this paper, one of the key advantages of K-GRN over other approaches lies in its ability to ensure trajectory traceability, thus bolstering drone safety. Concerning the time of generating potential field $t_{field}$, our method shows a minor drawback in terms of the duration needed for potential field formation. However, when evaluating the overall planning time, which encompasses both $t_{field}$ and $t_{plan}$, we still maintain a significant edge over alternative planning methods. Despite the slightly longer computational time required for potential field generation, the proposed method demonstrated superior performance in terms of the navigation task success rate and planning time compared with the other two methods. This highlights the effectiveness of the proposed approach in dynamic environments and its potential as an ideal choice for trajectory planning.

To evaluate the effectiveness of our trajectory planning framework, we compared it with two state-of-the-art methods: the Ego-planner [31] and the Fast-planner [28]. The Fast-Planner utilizes the ESDF for path search and incorporates the kinematic information in the search process, whereas the Ego-planner proposes a real-time planning method that does not rely on the ESDF. We conducted a series of experiments on six scenes with two obstacle-density levels. Both benchmark methods were open-source, and we used their default parameters for the experiments.

Table 3 presents a comparison between K-GRN and two state-of-the-art (SOTA) methods across two distinct scenarios. In this context, employing the same experimental methodology, we conducted separate comparisons of the drone’s navigation success rates, average flight speeds, and planned trajectory lengths under each condition.

Across all scenes, our proposed method significantly outperformed the two benchmark methods in terms of both success rates and average speeds. This demonstrates the effectiveness of the proposed approach in achieving successful navigation while maintaining rapid flight speed. However, our paths tended to be slightly longer than those generated by the other methods, due to our emphasis on flight safety. This trade-off ensures a higher level of safety during the flight.

Furthermore, Figure 4 displays the real-time speed variations among the three algorithms in dense scenes. It is worth highlighting that the presented algorithm showcases notably seamless speed transitions across the entire navigation journey. The pronounced speed reduction captured in the graph aligns with the drone executing obstacle avoidance maneuvers. While navigating through an identical course, our approach strategically adopts a safer path-planning strategy, prioritizing the avoidance of densely wooded narrow passages. Consequently, despite generating a lengthier route, this is a paramount benefit of our method.

To evaluate the practical efficacy of the K-GRN planner, we conducted relevant simulation experiments. The simulation environment was designed to replicate a real jungle scenario. In this scenario, we had the drone shuttles through the jungle. As shown in Figure 5, the UAV’s obstacle avoidance process, guided by the K-GRN planner, is depicted. It is clear that the drone could detect obstacles in advance, perform evasive maneuvers, and navigate successfully while maintaining a safe distance from the obstacles. Figure 6 displays the trajectory of the drone as it shuttles through the jungle environment.

In summary, our proposed method outperformed the benchmark methods in terms of success rate and average speed across various obstacle-density scenarios. The experimental results validated the superiority of the proposed approach, highlighting its potential to advance the field of UAV trajectory planning.

## 6. Conclusions

In this study, we have introduced an improved GRN potential field, termed the K-GRN potential field, specifically designed for trajectory planning. This potential field incorporates the costs of state transitions for intelligent agents to assist planners in generating smooth and efficient trajectories. Additionally, we have designed a planner that searches for the optimal trajectory using input sampling, requiring only a search for the path with the lowest cost within the established potential field. Through benchmark comparisons, we have validated the advantages of incorporating the state space in the GRN potential fields, the effectiveness of the K-GRN potential field in trajectory planning, and the performance of the K-GRN-guided planner compared with state-of-the-art methods. The results demonstrated that the proposed method offered superior paths in terms of both safety and effectiveness for drone flight.

Our approach also displays specific limitations: Driven by safety considerations, we embraced a cautious planning strategy, which results in a difficulty in devising optimal paths within complex scenarios. Similarly, due to this rationale, despite attaining advantages in average speed, our method encountered difficulties in outperforming the SOTA approach with regards to the highest planned speed. In forthcoming endeavors, we aim to enhance the K-GRN methodology by integrating adaptive planning strategies, thereby enabling it to demonstrate diverse obstacle avoidance performances across a spectrum of scenario complexities. Furthermore, we aim to extend the application of the K-GRN potential fields to drone swarm missions and explore their application in achieving cooperative flight in environments without communication capabilities.

## Figures and Tables

**Figure 1 sensors-23-07982-f001:**
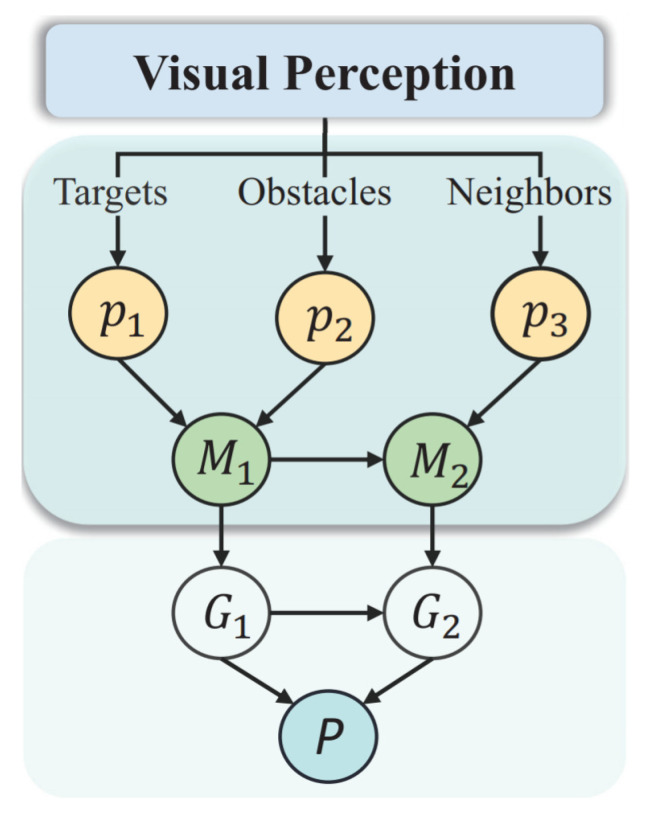
Schematic of the structure of the vision-based GRN. Each cell represents an individual UAV and is composed of an upper and lower layer. In the upper layer, sensory proteins $p_{1}$, $p_{2}$, and $p_{3}$ receive information about the positions of targets, obstacles, and neighboring robots, respectively, forming corresponding concentration fields. Protein $M_{1}$ integrates the concentration field from $p_{1}$ and $p_{2}$, while protein $M_{2}$ integrates the concentration field from $M_{1}$ and $p_{3}$. These proteins influence the production of actuating proteins $G_{1}$ and $G_{2}$ in the lower layer, which, respectively, represent entrapping patterns and the moving direction of the UAV. $G_{1}$ and $G_{2}$ also affect the production of protein *P*, which ultimately determines the dynamic position of the UAV. Moreover, protein *P* in one cell influences the gene expression of the neighboring cells in the system.

**Figure 2 sensors-23-07982-f002:**
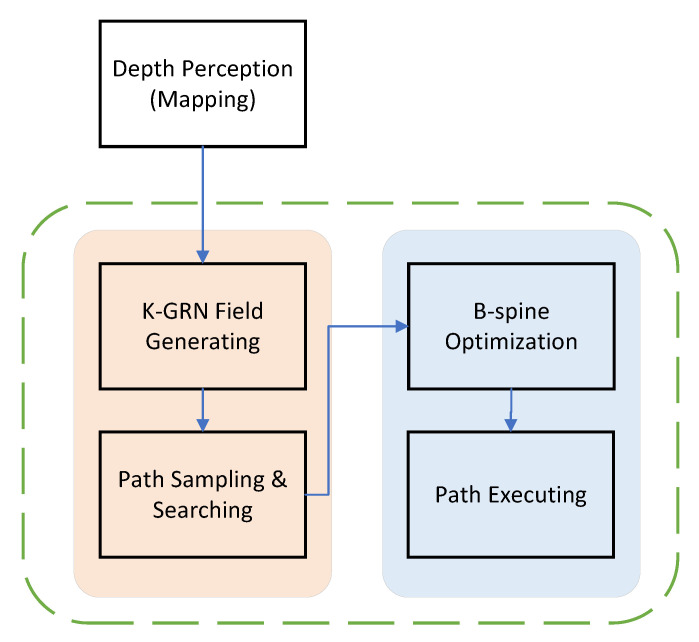
Flow diagram of the trajectory planner.

**Figure 3 sensors-23-07982-f003:**
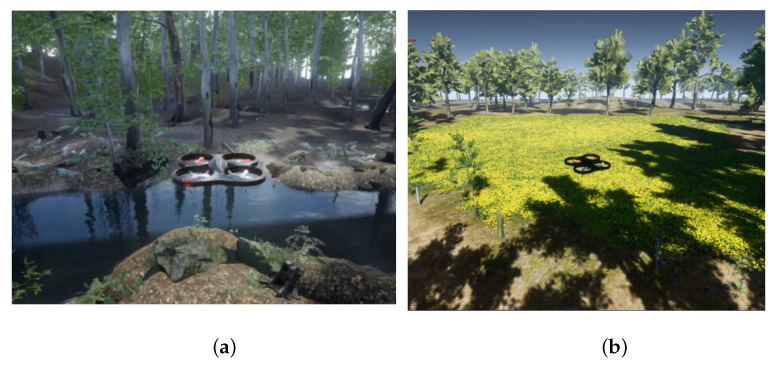
Illustrations of the two different scenes with varying obstacle densities. (**a**) Dense jungle scene. (**b**) Sparse jungle scene.

**Figure 4 sensors-23-07982-f004:**
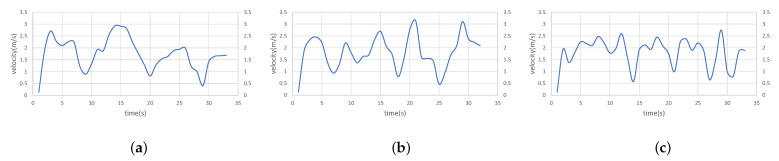
Real-time Velocity Variation of UAV during the Experiment: (**a**) Velocity Curve of GRN-planner. (**b**) Velocity Curve of Fast-planner. (**c**) Velocity Curve of Ego-planner.

**Figure 5 sensors-23-07982-f005:**
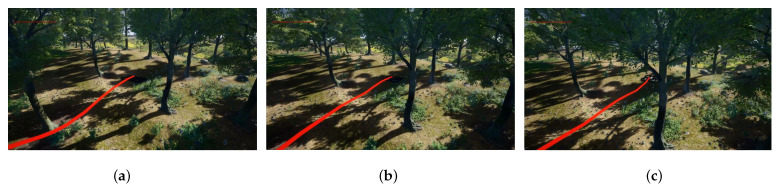
(**a**–**c**): Obstacle avoidance trajectories of a drone guided by the K-GRN planner in a jungle scenario.

**Figure 6 sensors-23-07982-f006:**
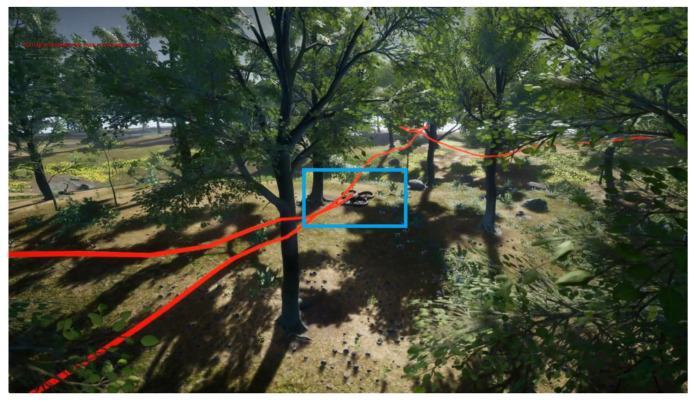
Trajectories of the drone as it shuttles through the jungle environment.

**Table 1 sensors-23-07982-t001:** Effect of Different Gradient Fields on Trajectory Generation.

Policy	Success Rate	Max Velocity (m/s)	Average Velocity (m/s)	Slove Time (ms)
EF	0.69	**4.13**	2.65	6.55
NF	0.64	3.81	2.47	**1.49**
GRN	**0.72**	3.93	**2.71**	6.88

The numbers in bold represent the optimal data for this column.

**Table 2 sensors-23-07982-t002:** Performance Comparison of Different Planning Methods After Incorporating UAV Kinematics Information into GRN Potential Field.

Policy	Success Rate	$t_{\mathrm{field}}$ (ms)	$t_{\mathrm{plan}}$ (ms)
KF	**0.72**	5.22	**0.91**
NKF	0.69	4.88	4.47
NF	0.55	**0**	2.90

The numbers in bold represent the optimal data for this column.

**Table 3 sensors-23-07982-t003:** Comparison of UAV Navigation Performance in Scenes with Different Obstacle Densities.

	Dense Scene			Sparse Scene		
**Policy**	**Success Rate**	**Average Velocity (m/s)**	**Trajectory Length (m)**	**Success Rate**	**Average Velocity (m/s)**	**Trajectory Length (m)**
GRN-planner	**0.56**	**2.01**	71.29	**0.72**	**2.71**	58.13
Fast-planner	0.47	1.95	**65.76**	0.69	2.65	55.43
Ego-planner	0.53	1.88	67.18	0.64	2.47	**49.90**

The numbers in bold represent the optimal data for this column.

## Data Availability

Not applicable.

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
