# Peer review of "Trajectory Planner for UAVs Based on Potential Field Obtained by a Kinodynamic Gene Regulation Network"

_sensors, 2023, doi:10.3390/s23187982_

Round 1

Reviewer 1 Report

This paper proposes a kind of path planning for UAVs based on Kinodynamic Gene Regulation Network Potential Field. In general, the research content of this paper is quite interesting and provides a reference method for UAVs path planning in complex environments. My comments are as follows:

(1) The research motivation of the paper needs to be more clear, and why K-GRN method should be used needs to be explained in detail, especially its advantages compared with other methods, and it is best to list this aspect.

(2) The method part is difficult to understand, there are too many formulas and parameters, it is best to add some more intuitive flow charts, so as to facilitate the understanding of the whole process of the algorithm.

(3) The content displayed in the experimental part is too little, and more pictures, data analysis results, tables, etc., should be placed.

(4) There is too little discussion about the experimental results in Table 2 and Table 3. The reasons why such results are achieved should be analyzed and summarized.

Moderate editing of English language is required.

Reviewer 2 Report

The authors develop a novel trajectory planner for UAVs based on Kinodynamic Gene Regulation Network. The planner is tested in real-life scenarios and compared with competitive solutions.

In the first part of the paper, the authors give an extensive literature review of the state of the art, highlighting their motivation to do the study. What I liked about this review is how the authors delve directly into specific groups of UAV control algorithms and analyze them. The second part of the paper explains the development process of their solution and experiment design, accompanied by the obtained results. The model is tested against the comparative planers and it showed better performance in most of the metrics used. The only downside of the proposed approach, compared to others, is the somewhat longer computation time needed for the trajectory computation. However, the authors explain this extra time as the time needed for finding a safe route, which makes sense.

I have several points that the authors should consider:

- Across the paper, there were typos (double space, or no space between the text and the colon).

- In Table 3, the trajectory length is measured in centimeters. Is that true? Should not that be meters? It seems odd to me that the drone traveled a trajectory measured in cm when its average speed over the 35 s was always >1.8 m/s.

- In the discussion of Figure 4, I would like to see the authors’ comments about the velocity changes in relation to the terrain and the trajectory taken by the drone.

- I would also like to see the authors’ view regarding how their solution might work in different use cases, e.g., if the drone would carry a load that would change its dimensions. Can your planner account for that change and adjust the trajectories accordingly?

- Adding to the previous point, add a discussion about the limitations of your solutions.

I don't think extensive proofreading is necessary, however, the typos need fixing. Carefully go through the text and correct them.

Round 2

Reviewer 1 Report

I suggest accept the paper.

English is OK.

Author Response

Dear Reviewer,

Thank you very much for your valuable comments. Your suggestions on how to improve the paper are greatly appreciated.

Best wishes,

Zhun Fan

on behalf of Wenji Li, Juncao Hong and Diquan Chen

Reviewer 2 Report

The authors addressed all my comments from the previous round so I have no further requirements.

I haven't noticed any errors in the latest version of the paper. I assume, during the production stages, if there are some minor errors (typos), those will be detected and fixed.

Author Response

Dear Academic Editor,

Thank you very much for your valuable comments. We have rechecked the typos issue and made the necessary corrections. Your suggestions on how to improve the paper are greatly appreciated.

Best wishes,  

Zhun Fan  

on behalf of Wenji Li, Juncao Hong and Diquan Chen